# GS-LiDAR: Generating Realistic LiDAR Point Clouds with Panoramic Gaussian Splatting

**Junzhe Jiang, Chun Gu, Yurui Chen, Li Zhang**[*]
School of Data Science, Fudan University

https://github.com/fudan-zvg/GS-LiDAR

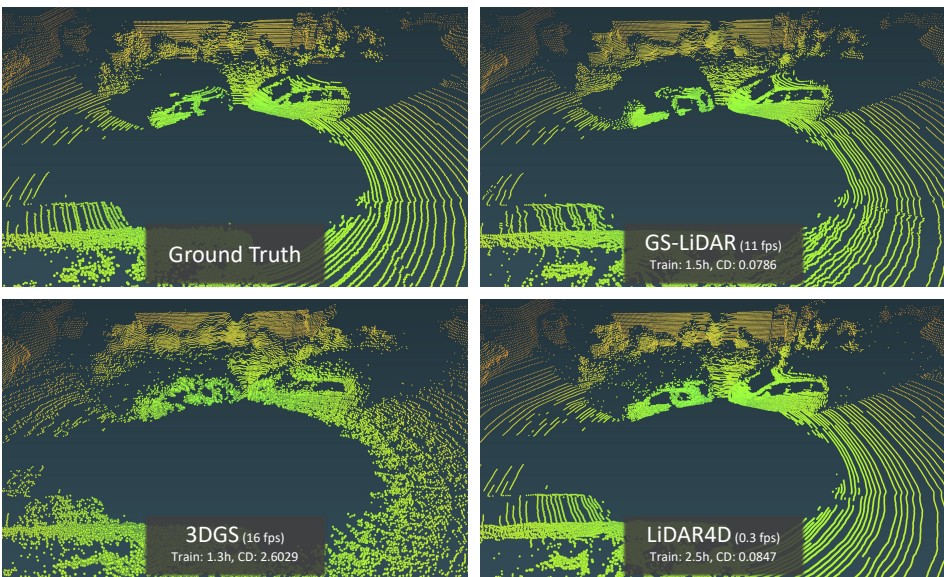

Figure 1: *GS-LiDAR* achieves superior LiDAR simulation quality for novel view synthesis while maintaining fast training and rendering speed.

## Abstract

LiDAR novel view synthesis (NVS) has emerged as a novel task within LiDAR simulation, offering valuable simulated point cloud data from novel viewpoints to aid in autonomous driving systems. However, existing LiDAR NVS methods typically rely on neural radiance fields (NeRF) as their 3D representation, which incurs significant computational costs in both training and rendering. Moreover, NeRF and its variants are designed for symmetrical scenes, making them ill-suited for driving scenarios. To address these challenges, we propose *GS-LiDAR*, a novel framework for generating realistic LiDAR point clouds with panoramic Gaussian splatting. Our approach employs 2D Gaussian primitives with periodic vibration properties, allowing for precise geometric reconstruction of both static and dynamic elements in driving scenarios. We further introduce a novel panoramic rendering technique with explicit ray-splat intersection, guided by panoramic LiDAR supervision. By incorporating intensity and ray-drop spherical harmonic (SH) coefficients into the Gaussian primitives, we enhance the realism of the rendered point clouds. Extensive experiments on KITTI-360 and nuScenes demonstrate the superiority of our method in terms of quantitative metrics, visual quality, as well as training and rendering efficiency.

## 1 Introduction

Captured data in driving scenarios is critically important, as it serves to train and simulate autonomous driving systems. However, the collection of driving data is both costly and inefficient.

---

[*]Corresponding author lizhangfd@fudan.edu.cn.

This underscores the necessity for a LiDAR simulation algorithm capable of generating realistic LiDAR data more efficiently. A common approach in the field is to reconstruct 3D street scenes from sparse data, which can then be used to generate novel view data. Most previous works (Xie et al., 2023b; Yang et al., 2024a; Chen et al., 2023a; Yan et al., 2024) have concentrated on novel view synthesis for cameras. These approaches use RGB images captured by vehicle-mounted cameras as input to reconstruct 3D scenes and render images from novel perspectives. Although significant progress has been made in camera-based novel view synthesis, the simulation of LiDAR remains underexplored. Due to the inherent sparsity of LiDAR point clouds, the challenge lies in accurately reconstructing 3D scenes using only LiDAR data. Furthermore, LiDAR sensors do not capture all emitted beams, as factors such as the reflective properties of objects affect beam reception, leading to point cloud dropout, which further increases the difficulty of 3D scene reconstruction.

Traditional LiDAR simulation methods can be broadly classified into two categories: virtual environment modeling (Dosovitskiy et al., 2017; Shah et al., 2018; Koenig & Howard, 2004) and reconstruction-based approaches (Manivasagam et al., 2020; Li et al., 2023; Guillard et al., 2022). The former involves creating 3D virtual worlds using physics engines and handcrafted 3D assets, though these simulators are constrained by the limited diversity and high cost of generating 3D assets. Additionally, there remains a significant domain gap between simulations and the real world. In contrast, reconstruction-based methods aim to address these limitations by reconstructing 3D assets from real-world captured data. However, the complexity of multi-step algorithms in this category limits their practical application in industry.

In recent years, neural radiance field (NeRF) (Mildenhall et al., 2020) has emerged as a foundational technique in the field of 3D reconstruction. With its effective implicit representation and high-quality volumetric rendering, NeRF has been proposed as a novel approach for LiDAR simulation. NeRF-LiDAR (Zhang et al., 2024) utilizes real images and LiDAR data to learn a Neural Radiance Field, generating point clouds and rendering semantic labels. On the other hand, LiDAR-NeRF (Tao et al., 2023) introduces a novel LiDAR view synthesis task, which uses only LiDAR data as input to reconstruct a 3D scene. However, LiDAR-NeRF is limited to modeling static scenes, whereas dynamic vehicles and pedestrians are common in driving scenarios. To address this, LiDAR4D (Zheng et al., 2024) introduces a hybrid 4D representation for novel space-time LiDAR view synthesis. Although these NeRF-based methods mark significant progress compared to traditional techniques, they suffer from slow training and rendering speeds, a limitation inherent to NeRF. Furthermore, efficient NeRF-based architectures like HashGrid (Müller et al., 2022), designed for symmetrical scenes, are not well-suited for driving scenarios.

In this paper, we propose **GS-LiDAR**, a novel framework for generating realistic LiDAR point clouds using panoramic Gaussian splatting. While 3D Gaussian splatting struggles with geometric modeling and tends to overfit sparse views, we leverage view-consistent 2D Gaussian primitives Huang et al. (2024) for more accurate geometric representation. Moreover, considering the dynamic nature of driving scenarios, we introduce periodic vibration properties Chen et al. (2023a) into the Gaussian primitives, enabling the uniform representation of various objects and elements in dynamic environments. Focusing on the task of novel LiDAR view synthesis, we introduce a novel panoramic rendering process to facilitate fast and efficient rendering of panoramic depth maps using 2D Gaussian primitives. Through carefully designed ray-splat intersections, the resulting panoramic depth maps are geometrically accurate and view-consistent. Each Gaussian is assigned additional LiDAR-specific attributes, such as view-dependent intensity and ray-drop probability, which are aggregated into intensity maps and ray-drop maps through alpha-blending in the splatting process. By using ground-truth LiDAR range maps and intensity maps for supervision, *GS-LiDAR* can effectively simulate LiDAR point clouds. As illustrated in Figure 1, *GS-LiDAR* achieves superior LiDAR simulation quality in novel LiDAR view synthesis and significantly outperforms the previous state-of-the-art method, LiDAR4D (Zheng et al., 2024), in both training and rendering speed.

We conduct extensive experiments to evaluate the effectiveness of our method on two major benchmarks: KITTI-360 (Liao et al., 2022) and nuScenes (Caesar et al., 2020). For static scenes, we test our approach on the KITTI-360 dataset, achieving a substantial 10.7% reduction in the RMSE metric compared to the leading competitor, LiDAR4D (Zheng et al., 2024). For dynamic scenes, our method outperforms LiDAR4D, with RMSE reductions of 11.5% on KITTI-360 and 13.1% on nuScenes. Moreover, our approach demonstrated significantly faster training and rendering times than previous NeRF-based methods, with a speedup of 1.67 times in training and a notable increase of 31 times in rendering LiDAR novel views compared to LiDAR4D.

Our **contributions** are summarized as follows: **(1)** We propose *GS-LiDAR*, a novel differentiable framework for generating realistic LiDAR point clouds. **(2)** We employ 2D Gaussian primitives with periodic vibration properties, enabling precise geometric reconstruction of various objects and elements in dynamic scenarios. **(3)** We introduce a novel panoramic rendering technique based on 2D Gaussian primitives, with geometrically accurate ray-splat intersection, where the rendered panoramic maps are supervised by the ground-truth data. **(4)** Extensive experiments demonstrate the superiority of our method across quantitative metrics, visual quality, as well as training and rendering speeds, when compared to previous approaches.

## 2 RELATED WORK

**Novel view synthesis** Novel view synthesis (NVS) is a critical and challenging aspect of 3D reconstruction. Since the advent of neural radiance fields (NeRF) (Mildenhall et al., 2020), there have been significant advancements in 3D reconstruction and NVS. NeRF utilizes a multi-layer perceptron (MLP) to model geometric shapes and view-dependent appearances, rendering through volume rendering. NeRF has demonstrated that implicit radiance fields can effectively learn scene representations and synthesize high-quality novel views. Despite its profound impact, NeRF faces notable challenges, including slow rendering speeds and aliasing. To address these issues, various research efforts (Barron et al., 2021; 2022; 2023; Hu et al., 2023; Reiser et al., 2021; Yu et al., 2021; Reiser et al., 2023; Hedman et al., 2021; Yariv et al., 2023; Chen et al., 2023c; 2022; Müller et al., 2022; Liu et al., 2020; Sun et al., 2022; Chen et al., 2023b; Fridovich-Keil et al., 2022) have developed variants that focus on enhancing rendering quality as well as accelerating training and rendering speeds. Recently, 3D Gaussian splatting (3DGS) (Kerbl et al., 2023) has introduced a point-based 3D scene representation that innovatively combines high-quality alpha-blending with rapid rasterization. 3DGS has been swiftly extended to various domains to enhance its rendering capabilities (Xie et al., 2023a; Huang et al., 2024; Gao et al., 2024) and its potential to represent dynamic scenes (Yang et al., 2024b; Yan et al., 2024; Zielonka et al., 2023; Chen et al., 2023a). In this paper, we adopt 2D Gaussian primitives with periodic vibration properties as scene representation to characterize the accurate geometry of both static and dynamic elements.

**LiDAR simulation** Traditional simulators can be classified into two categories. The first type of method (Dosovitskiy et al., 2017; Shah et al., 2018; Koenig & Howard, 2004) utilizes physics engines to generate LiDAR point clouds through ray casting within handcrafted virtual environments. However, these simulators are limited by their diversity and the high cost of 3D assets, and they exhibit a significant domain gap when compared to real-world data. In contrast, the second type of methods (Manivasagam et al., 2020; Li et al., 2023; Guillard et al., 2022) have sought to address these limitations by reconstructing scenes from real data for simulation. For instance, LiDARsim (Manivasagam et al., 2020) and PCGen (Li et al., 2023) utilize a multi-step, data-driven approach to simulate point clouds from real data. However, the complexity of these multiple steps impacts their applicability and scalability. More recent works (Tao et al., 2023; Zhang et al., 2024; Zheng et al., 2024; Xue et al., 2024; Tao et al., 2024; Wu et al., 2024) have utilized NeRF for scene reconstruction and LiDAR simulation, achieving higher quality and more reliable results. However, NeRF is time-consuming due to its implicit representation and exhaustive ray marching. Moreover, NeRF and its variants are mostly designed for symmetrical scenes, making them ill-suited to the driving scenarios. To this end, *GS-LiDAR* employs the explicit representation of Gaussian splatting, which enables efficient and flexible LiDAR simulation.

## 3 METHOD

In this section, we propose *GS-LiDAR*, a novel framework for generating realistic LiDAR point clouds with Gaussian splatting. We begin by introducing the necessary background on 3D Gaussian splatting in Section 3.1. For geometrically accurate reconstruction and the modeling of both static and dynamic elements, we employ 2D Gaussian primitives with periodic vibration properties as our scene representation, as outlined in Section 3.2. To integrate LiDAR supervision, we propose an innovative panoramic rendering technique with explicit ray-splat intersection, described in Section 3.3. Next, we detail the LiDAR modeling approach, including the rendering of depth maps,

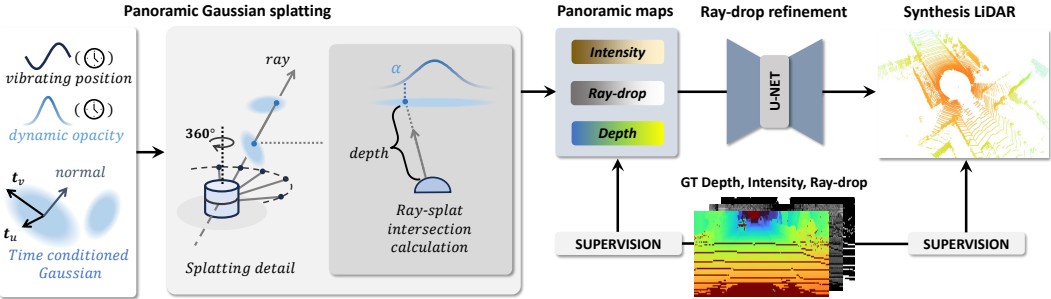

Figure 2: Overview of the *GS-LiDAR* framework: *GS-LiDAR* is based on 2D Gaussian primitives with periodic vibration properties, allowing for dynamic modeling of position and opacity along with accurate geometry. At a given timestamp, Gaussians query their states and utilize the proposed panoramic Gaussian splatting technique to render panoramic maps of depth, ray-drop, and intensity. For each ray and Gaussian primitive, we calculate their intersection to obtain the depth and $\alpha$, ensuring more geometrically consistent renderings. The results are subsequently refined by a well-trained U-Net to further enhance the quality of the point clouds.

intensity maps, and ray-drop probability maps, in Section 3.4. Finally, in Section 3.5, we discuss the various functions used to optimize the scenes. An overview of our pipeline is provided in Figure 2.

## 3.1 PRELIMINARY: 3D GAUSSIAN SPLATTING

3D Gaussian splatting (3DGS) (Kerbl et al., 2023) employs a set of anisotropic Gaussian primitives to represent a static 3D scene, which is subsequently rendered vis differentiable splatting. By utilizing a tile-based rasterizer, this approach facilitates the real-time rendering of novel views with high visual fidelity. Each Gaussian primitive is characterized by a position vector $\boldsymbol{\mu} \in \mathbb{R}^3$, a covariance matrix $\boldsymbol{\Sigma} \in \mathbb{R}^{3 \times 3}$, an opacity parameter $o \in (0, 1)$, and color $\boldsymbol{c} \in \mathbb{R}^3$ modeled by spherical harmonics (SH). The influence $\mathcal{G}(\boldsymbol{x})$ of a given Gaussian primitive on a spatial position $\boldsymbol{x}$ is defined by an unnormalized Gaussian function:

$$\mathcal{G}(\boldsymbol{x}) = \exp(-\frac{1}{2}(\boldsymbol{x} - \boldsymbol{\mu})^\top \boldsymbol{\Sigma}^{-1}(\boldsymbol{x} - \boldsymbol{\mu})). \tag{1}$$

To render an image, the 3D Gaussian primitives are initially transformed into the camera coordinate system via the view transformation matrix $\boldsymbol{W}$. Following this transformation, each Gaussian primitive is projected onto the image plane through a local affine transformation $\boldsymbol{J}$, which maps the 3D structure into 2D image space. The 2D covariance matrix $\boldsymbol{\Sigma}'$ of the projected Gaussian primitive in camera coordinates is computed as: $\boldsymbol{\Sigma}' = \boldsymbol{J} \boldsymbol{W} \boldsymbol{\Sigma} \boldsymbol{W}^\top \boldsymbol{J}^\top$. The final rendering process employs alpha-blending, wherein the color $\boldsymbol{c}$ of a target pixel is obtained by aggregating the contribution $\mathcal{G}'$ of each relevant projected 2D Gaussian, proceeding from front to back:

$$\boldsymbol{c} = \sum_{k=1}^{K} \boldsymbol{c}_k \, o_k \, \mathcal{G}'_k \prod_{j=1}^{k-1} (1 - o_j \, \mathcal{G}'_j), \tag{2}$$

where $k$ denotes the index of a Gaussian primitive, with $K$ representing the total number of Gaussian primitives in the scene.

## 3.2 PERIODIC VIBRATION 2D GAUSSIAN

Given the constant presence of moving vehicles and pedestrians in driving scenarios, we aim to utilize a unified representation to model the various objects and elements within the scene. We employ 2D Gaussian primitives (Huang et al., 2024) with periodic vibration properties (Chen et al., 2023a) to accurately capture surface behavior across both space and time. For a 2D Gaussian defined by its central point $\boldsymbol{\mu} \in \mathbb{R}^3$, an opacity parameter $o \in [0, 1]$, two principal tangential vectors $\boldsymbol{t}_u \in \mathbb{R}^3$ and $\boldsymbol{t}_v \in \mathbb{R}^3$, and a scaling vector $\boldsymbol{s} = (s_u, s_v) \in \mathbb{R}^2$, we introduce additional learnable attributes that govern the variation of its central point and opacity. These include the vibrating direction

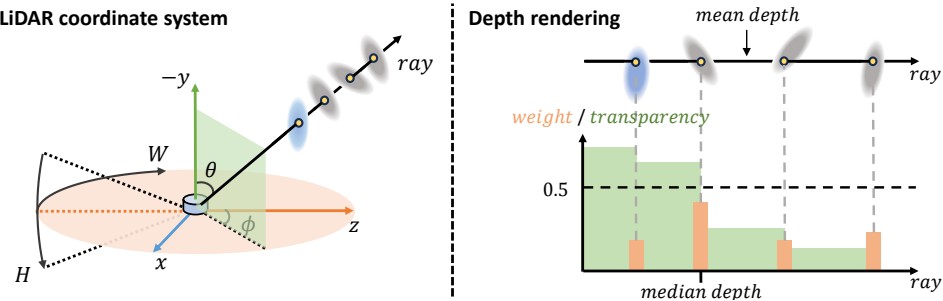

Figure 3: Our LiDAR coordinate system and two ways of depth rendering. The mean depth refers to the weighted average of each depth using the rendering weights, while the median depth is defined as the maximum depth with transparency, i.e., $\prod_{j=1}^{k-1}(1 - o_j \mathcal{G}_j)$, no greater than 0.5.

$\boldsymbol{v} \in \mathbb{R}^3$, life peak $\tau \in \mathbb{R}$, and time decay rate $\beta \in \mathbb{R}$:

$$\tilde{\boldsymbol{\mu}}(t) = \boldsymbol{\mu} + \frac{l}{2\pi} \cdot \sin(\frac{2\pi(t - \tau)}{l}) \cdot \boldsymbol{v}, \tag{3}$$

$$\tilde{o}(t) = o \cdot \exp(-\frac{1}{2}(t - \tau)^2 \beta^{-2}), \tag{4}$$

where $\tilde{\boldsymbol{\mu}}(t)$ represents the vibrating motion around $\boldsymbol{\mu}$ and $\tau$, and $\tilde{o}(t)$ represents the decayed opacity. The hyper-parameter $l$ denotes the cycle length, which serves as a prior for the scene. Each equipped with a simple motion, the Gaussian primitives can join up to represent any complex motion of dynamic elements in a relay manner. The influence $\mathcal{G}$ of each 2D Gaussian disk is defined within its local tangent plane in world space:

$$\mathcal{G}(u, v) = \exp\left(-\frac{u^2 + v^2}{2}\right), \tag{5}$$

where $(u, v)$ are the coordinates within the local tangent plane space (UV space). The transformation from UV space to screen space is parameterized as:

$$\boldsymbol{x}(u, v) = \boldsymbol{W}(\tilde{\boldsymbol{\mu}}(t) + s_u \boldsymbol{t}_u u + s_v \boldsymbol{t}_v v) = \boldsymbol{W}\boldsymbol{H}(u, v, 1)^\top, \tag{6}$$

$$\text{where } \boldsymbol{H} = \begin{bmatrix} s_u \boldsymbol{t}_u & s_v \boldsymbol{t}_v & \tilde{\boldsymbol{\mu}}(t) \\ 0 & 0 & 1 \end{bmatrix} \in \mathbb{R}^{4 \times 3}. \tag{7}$$

Here, $\boldsymbol{W} \in \mathbb{R}^{4 \times 4}$ represents the view transformation matrix, and $\boldsymbol{x} = (x, y, z, 1)^\top$ denotes the homogeneous coordinates in sensor space.

### 3.3 PANORAMIC GAUSSIAN SPLATTING

LiDAR emits laser pulses and measures the time-of-flight (ToF) to determine object distances, as well as the intensity of reflected light. Spinning LiDAR provides a 360-degree horizontal field of view $(-\pi, \pi)$ and a limited vertical field of view $(\text{VFOV}_{\text{min}}, \text{VFOV}_{\text{max}})$, allowing it to perceive the environment with a specific angular resolution. The angular configuration within the LiDAR coordinate system is depicted in Figure 3. Given the pixel coordinates of a point on the range image $(\xi, \eta)$, the corresponding radian angles can be computed using the following equation:

$$\begin{pmatrix} \phi \\ \theta \end{pmatrix} = \begin{pmatrix} (2\xi - W)\pi W^{-1} \\ \eta H^{-1}(\text{VFOV}_{\text{max}} - \text{VFOV}_{\text{min}}) + \text{VFOV}_{\text{min}} \end{pmatrix}, \tag{8}$$

where $(W, H)$ represent the width and height of the range image, respectively. A homogeneous point in the LiDAR coordinate system can then be determined from $(\phi, \theta)$ as follows: $\boldsymbol{x} = (x, y, z, 1)^\top = (r \sin\theta \sin\phi, -r \cos\theta, r \sin\theta \cos\phi, 1)^\top$, where $r = \sqrt{x^2 + y^2 + z^2}$ denotes the distance from the point to the center of the LiDAR.

**Ray-splat intersection** We define the ray as the intersection of two orthogonal homogeneous planes, $\boldsymbol{h}_x = (\cos\phi, 0, -\sin\phi, 0)^\top$ and $\boldsymbol{h}_y = (\cos\theta \sin\phi, \sin\theta, \cos\theta \cos\phi, 0)^\top$, characterized by their normal vectors. These satisfy the conditions $\boldsymbol{h}_x^\top \boldsymbol{x} = 0$ and $\boldsymbol{h}_y^\top \boldsymbol{x} = 0$. Consequently, given

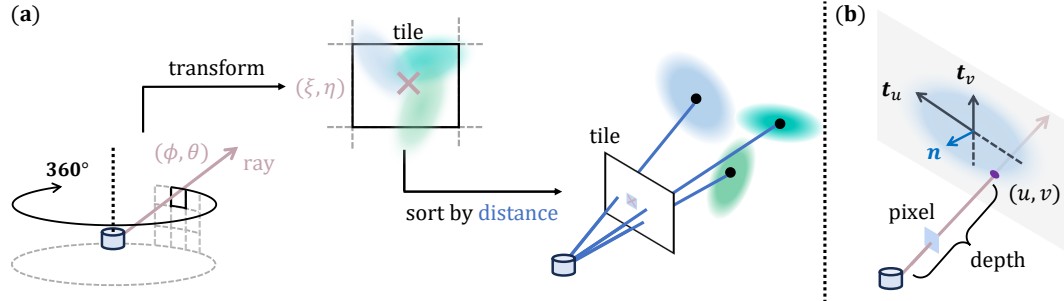

Figure 4: Panoramic Gaussian rasterization details. **(a)** Our method employs tile-based sorting and rendering. For panoramic ray maps, we first transform the epipolar coordinate system into the pixel coordinate system. The pixel map is then divided into small tiles, and within each tile, Gaussian primitives are sorted based on their distance to the LiDAR origin. **(b)** During pixel rendering, the $\alpha$ and depth are computed by calculating the intersection between the ray and the Gaussian primitive.

the ray angles $(\phi, \theta)$, the ray-splat intersection $(u, v)$ is determined by solving the following linear equations:

$$[\boldsymbol{h}_x, \boldsymbol{h}_y]^\top \boldsymbol{W} \boldsymbol{H}(u, v, 1)^\top = 0. \tag{9}$$

Let $[\boldsymbol{R}, \boldsymbol{h}] = [\boldsymbol{h}_x, \boldsymbol{h}_y]^\top \boldsymbol{W} \boldsymbol{H}$, where $\boldsymbol{R} \in \mathbb{R}^{2\times2}$ and $\boldsymbol{h} \in \mathbb{R}^2$, From this, we can solve for $u$ and $v$ as follows:

$$\begin{pmatrix} u \\ v \end{pmatrix} = -\boldsymbol{R}^{-1}\boldsymbol{h}. \tag{10}$$

## 3.4 LiDAR RENDERING

In the novel LiDAR view synthesis task, LiDAR point clouds, which include intensity values, can be projected to panoramic range maps and intensity maps. To simulate LiDAR point clouds, we assign each Gaussian a view-dependent intensity value $\lambda$ and a view-dependent ray-drop probability $\rho$, both of which are modeled using spherical harmonics.

**Depth map** Considering the ray-splat intersection within the LiDAR coordinate system, the depth value corresponds to the distance $r$ from the intersection point to the center of the LiDAR. Based on Equation 7 and the conversion between $(x, y, z)$ and $(\phi, \theta)$, we have:

$$(r \sin\theta \sin\phi, -r \cos\theta, r \sin\theta \cos\phi, 1)^\top = \boldsymbol{W} \boldsymbol{H}(u, v, 1)^\top. \tag{11}$$

By multiplying $(\sin\theta \sin\phi, -\cos\theta, \sin\theta \cos\phi, 0)$ on both sides, the distance $r$ can be computed as:

$$r = (\sin\theta \sin\phi, -\cos\theta, \sin\theta \cos\phi, 0)\boldsymbol{W} \boldsymbol{H}(u, v, 1)^\top. \tag{12}$$

Although we calculate the intersection of each ray with the Gaussian primitives, we still adopt the tile-based rendering method proposed in 3D Gaussian splatting Kerbl et al. (2023) to achieve efficient rendering. As illustrated in Figure 4, we first transform the epipolar coordinates into pixel coordinates and sort the Gaussian primitives within each tile. For pixel rendering, the $\alpha$ and depth are determined based on the intersection results.

During the training process, we utilize both the mean depth and the median depth, and supervise them using the projected ground truth range map as follows:

$$R_{\text{mean}} = \sum_{k=1}^{K} r_k \, o_k \, \mathcal{G}_k \prod_{j=1}^{k-1} (1 - o_j \, \mathcal{G}_j), \tag{13}$$

$$R_{\text{median}} = \max \left\{ r_k \,\middle|\, \prod_{j=1}^{k-1} (1 - o_j \, \mathcal{G}_j) > 0.5 \right\}, \tag{14}$$

$$\mathcal{L}_{\text{d}} = \| R_{\text{mean}} - R_{\text{gt}} \|_1 + \| R_{\text{median}} - R_{\text{gt}} \|_1 . \tag{15}$$

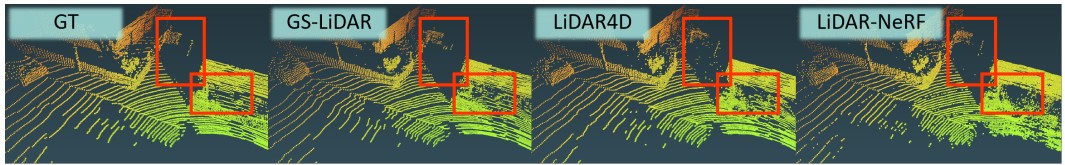

Figure 5: Comparison of 3D LiDAR point cloud. *GS-LiDAR* produces a more cohesive LiDAR point cloud compared to LiDAR-NeRF (Tao et al., 2023) and LiDAR4D (Zheng et al., 2024).

Table 1: **State-of-the-art comparison on KITTI-360 *Static* Scene Sequence**. We color the top results as best and second best .

| Method | Point Cloud | | Depth | | | | | Intensity | | | | |
|---|---|---|---|---|---|---|---|---|---|---|---|---|
| | CD↓ | F-score↑ | RMSE↓ | MedAE↓ | LPIPS↓ | SSIM↑ | PSNR↑ | RMSE↓ | MedAE↓ | LPIPS↓ | SSIM↑ | PSNR↑ |
| LiDARsim (Manivasagam et al., 2020) | 2.2249 | 0.8667 | 6.5470 | 0.0759 | 0.2289 | 0.7157 | 21.7746 | 0.1532 | 0.0506 | 0.2502 | 0.4479 | 16.3045 |
| NKSR (Huang et al., 2023) | 0.5780 | 0.8685 | 4.6647 | 0.0698 | 0.2295 | 0.7052 | 22.5390 | 0.1565 | 0.0536 | 0.2429 | 0.4200 | 16.1159 |
| PCGen (Li et al., 2023) | 0.2090 | 0.8597 | 4.8838 | 0.1785 | 0.5210 | 0.5062 | 24.3050 | 0.2005 | 0.0818 | 0.6100 | 0.1248 | 13.9606 |
| LiDAR-NeRF (Tao et al., 2023) | 0.0923 | 0.9226 | 3.6801 | 0.0667 | 0.3523 | 0.6043 | 26.7663 | 0.1557 | 0.0549 | 0.4212 | 0.2768 | 16.1683 |
| LiDAR4D (Zheng et al., 2024) | 0.0894 | 0.9264 | 3.2370 | 0.0507 | 0.1313 | 0.7218 | 27.8840 | 0.1343 | 0.0404 | 0.2127 | 0.4698 | 17.4529 |
| **GS-LiDAR (Ours)** | 0.0847 | 0.9236 | 2.8895 | 0.0411 | 0.0997 | 0.8454 | 28.8807 | 0.1211 | 0.0359 | 0.1630 | 0.5756 | 18.3506 |

**Intensity map** LiDAR intensity refers to the strength of the laser pulse upon its return to the receiver, which is influenced by material properties, surface characteristics, distance, and the incident angle. Intensity data contributes to terrain analysis, facilitates the differentiation of various features, and enhances the classification and precision of point cloud data. Similar to the rendering of mean depth, we aggregate the intensity map $I$ using alpha-blending: $I = \sum_{k=1}^{K} \lambda_k \, o_k \, \mathcal{G}_k \prod_{j=1}^{k-1}(1 - o_j \, \mathcal{G}_j)$, which is subsequently supervised by the ground truth intensity:

$$\mathcal{L}_{\mathrm{int}} = \|I - I_{\mathrm{gt}}\|_1. \tag{16}$$

**Ray-drop probability map** LiDAR ray-drop refers to the phenomenon where some laser pulses do not return to the sensor after striking surfaces, often due to obstructions, absorption by vegetation, or unfavorable angles of incidence. Through alpha-blending, we get a ray-drop probability map from Gaussians: $P_{\mathrm{gs}} = \sum_{k=1}^{K} \rho_k \, o_k \, \mathcal{G}_k \prod_{j=1}^{k-1}(1 - o_j \, \mathcal{G}_j)$. Additionally, since the LiDAR ray-drop is also related to the characteristics of the LiDAR itself, we introduce a learnable prior $P_{\mathrm{prior}}$ for the ray-drop. The final ray-drop probability map is expressed as: $P = P_{\mathrm{prior}} + (1 - P_{\mathrm{prior}})P_{\mathrm{gs}}$, which is supervised by the ground truth mask using a binary cross-entropy loss:

$$\mathcal{L}_{\mathrm{drop}} = \mathrm{BCE}\,(P, P_{\mathrm{gt}}). \tag{17}$$

However, this modeling of ray-drop does not account for factors such as distance to the sensor, potentially leading to unreliable results. To mitigate this issue, following NeRF-LiDAR (Zhang et al., 2024), we utilize a U-Net (Ronneberger et al., 2015) with residual connections to globally refine the ray-drop mask, thereby preserving consistent patterns across regions. Specifically, the U-Net takes the rendered ray-drop probability map $P$, depth map $R_{\mathrm{mean}}$, and intensity map $I$ as inputs, and outputs the refined ray-drop mask $P_{\mathrm{unet}}$. After training the Gaussians, we continue optimizing the U-Net by supervising the refined ray-drop mask using the same loss function as in Equation 17.

### 3.5 Loss function

To achieve accurate geometry and align the 2D splats with surfaces, we integrate the depth distortion loss and normal consistency loss from 2DGS Huang et al. (2024). Additionally, we employ the chamfer distance loss (Fan et al., 2017) to minimize the disparity between our simulated LiDAR point clouds and the ground truth data.

**Depth distortion** The depth distortion loss aims to reduce the distance between ray-splat intersections, encouraging the 2D Gaussian primitives to concentrate on the surface:

$$\mathcal{L}_{\mathrm{dist}} = \sum_{i,j} \omega_i \omega_j (r_i - r_j)^2, \tag{18}$$

where $\omega_i$ represents the blending weight of the $i$-th intersection, and $r_i$ denotes the depth of the intersection points.

Table 2: **State-of-the-art comparison on KITTI-360 dataset**. We color the top results as best and second best .

| Method | Point Cloud | | Depth | | | | | Intensity | | | | |
|---|---|---|---|---|---|---|---|---|---|---|---|---|
| | CD↓ | F-score↑ | RMSE↓ | MedAE↓ | LPIPS↓ | SSIM↑ | PSNR↑ | RMSE↓ | MedAE↓ | LPIPS↓ | SSIM↑ | PSNR↑ |
| LiDARsim (Manivasagam et al., 2020) | 3.2228 | 0.7157 | 6.9153 | 0.1279 | 0.2926 | 0.6342 | 21.4608 | 0.1666 | 0.0569 | 0.3276 | 0.3502 | 15.5853 |
| NKSR (Huang et al., 2023) | 1.8982 | 0.6855 | 5.8403 | 0.0996 | 0.2752 | 0.6409 | 23.0368 | 0.1742 | 0.0590 | 0.3337 | 0.3517 | 15.2081 |
| PCGen (Li et al., 2023) | 0.4636 | 0.8023 | 5.6583 | 0.2040 | 0.5391 | 0.4903 | 23.1675 | 0.1970 | 0.0763 | 0.5926 | 0.1351 | 14.1181 |
| LiDAR-NeRF (Tao et al., 2023) | 0.1438 | 0.9091 | 4.1753 | 0.0566 | 0.2797 | 0.6568 | 25.9878 | 0.1404 | 0.0443 | 0.3135 | 0.3831 | 17.1549 |
| D-NeRF (Pumarola et al., 2021) | 0.1442 | 0.9128 | 4.0194 | 0.0508 | 0.3061 | 0.6634 | 26.2344 | 0.1369 | 0.0440 | 0.3409 | 0.3748 | 17.3554 |
| TiNeuVox-B (Fang et al., 2022) | 0.1748 | 0.9059 | 4.1284 | 0.0502 | 0.3427 | 0.6514 | 26.0267 | 0.1363 | 0.0453 | 0.4365 | 0.3457 | 17.3535 |
| K-Planes (Fridovich-Keil et al., 2023) | 0.1302 | 0.9123 | 4.1322 | 0.0539 | 0.3457 | 0.6385 | 26.0236 | 0.1415 | 0.0498 | 0.4081 | 0.3008 | 17.0167 |
| LiDAR4D (Zheng et al., 2024) | 0.1089 | 0.9272 | 3.5256 | 0.0404 | 0.1051 | 0.7647 | 27.4767 | 0.1195 | 0.0327 | 0.1845 | 0.5304 | 18.5561 |
| **GS-LiDAR (Ours)** | **0.1085** | 0.9231 | **3.1212** | **0.0340** | **0.0902** | **0.8553** | **28.4381** | **0.1161** | **0.0313** | **0.1825** | **0.5914** | **18.7482** |

Table 3: **State-of-the-art comparison on nuScenes dataset**. The notations are consistent with the KITTI-360 Table 2 above.

| Method | Point Cloud | | Depth | | | | | Intensity | | | | |
|---|---|---|---|---|---|---|---|---|---|---|---|---|
| | CD↓ | F-score↑ | RMSE↓ | MedAE↓ | LPIPS↓ | SSIM↑ | PSNR↑ | RMSE↓ | MedAE↓ | LPIPS↓ | SSIM↑ | PSNR↑ |
| LiDARsim Manivasagam et al. (2020) | 12.1383 | 0.6512 | 10.5539 | 0.3572 | 0.1871 | 0.5653 | 17.7841 | 0.0659 | 0.0115 | 0.1160 | 0.5170 | 23.7791 |
| NKSR Huang et al. (2023) | 11.4910 | 0.6178 | 9.3731 | 0.5763 | 0.2111 | 0.5637 | 18.7774 | 0.0680 | 0.0119 | 0.1290 | 0.5031 | 23.4905 |
| PCGen Li et al. (2023) | 2.1998 | 0.6341 | 8.8364 | 0.4011 | 0.1792 | 0.5440 | 19.2799 | 0.0768 | 0.0147 | 0.1308 | 0.4410 | 22.4428 |
| LiDAR-NeRF Tao et al. (2023) | 0.3225 | 0.8576 | 7.1566 | 0.0338 | 0.0702 | 0.7188 | 21.2129 | 0.0467 | 0.0076 | 0.0483 | 0.7264 | 26.9927 |
| D-NeRF Pumarola et al. (2021) | 0.3296 | 0.8513 | 7.1089 | 0.0368 | 0.0789 | 0.7130 | 21.2594 | 0.0467 | 0.0080 | 0.0492 | 0.7180 | 26.9951 |
| TiNeuVox-B Fang et al. (2022) | 0.3920 | 0.8627 | 7.2093 | 0.0290 | 0.1549 | 0.6873 | 21.0932 | 0.0462 | 0.0080 | 0.1294 | 0.7107 | 26.8620 |
| K-Planes Fridovich-Keil et al. (2023) | 0.2982 | 0.8887 | 6.7960 | 0.0209 | 0.1218 | 0.7258 | 21.6203 | 0.0438 | 0.0076 | 0.1127 | 0.7364 | 27.4227 |
| LiDAR4D (Zheng et al., 2024) | 0.2443 | 0.8915 | 6.7831 | 0.0258 | 0.0569 | 0.7396 | 21.7189 | 0.0426 | 0.0071 | 0.0459 | 0.7498 | 27.7977 |
| **GS-LiDAR (Ours)** | **0.2382** | **0.9055** | **5.8925** | **0.0198** | 0.0708 | **0.8394** | **22.2482** | **0.0415** | **0.0067** | 0.0627 | 0.7291 | 27.7420 |

**Normal consistency**   Since our approach is based on 2D Gaussian primitives, it is crucial to ensure that all 2D splats are locally aligned with the actual surfaces. To achieve this, we align the rendered normal map $N$ with the pseudo-normal map $\tilde{N}$, which is computed from the gradients of the depth maps:

$$\mathcal{L}_n = 1 - N^\top \tilde{N}. \tag{19}$$

**Chamfer distance loss**   We also incorporate chamfer distance to introduce explicit geometric constraints from the input LiDAR point clouds. By back-projecting both the rendered and ground truth range images into 3D point clouds, $S_{\text{render}}$ and $S_{\text{gt}}$, respectively, we minimize the distance between the two point clouds using the chamfer distance (Fan et al., 2017):

$$\mathcal{L}_{\text{ch}} = \text{CD}(S_{\text{render}}, S_{\text{gt}}). \tag{20}$$

The overall loss function is defined as:

$$\mathcal{L} = \lambda_d \mathcal{L}_d + \lambda_{\text{int}} \mathcal{L}_{\text{int}} + \lambda_{\text{drop}} \mathcal{L}_{\text{drop}} + \lambda_{\text{dist}} \mathcal{L}_{\text{dist}} + \lambda_n \mathcal{L}_n + \lambda_{\text{ch}} \mathcal{L}_{\text{ch}}. \tag{21}$$

# 4 EXPERIMENT

## 4.1 EXPERIMENT SETUP

**Datasets**   We conduct extensive experiments on both dynamic and static scenes using the KITTI-360 (Liao et al., 2022) and nuScenes (Caesar et al., 2020) datasets, with the dynamic scenes featuring a significant number of moving vehicles. The KITTI-360 dataset employs a 64-beam LiDAR with a vertical field of view (FOV) of 26.4 degrees and an acquisition frequency of 10Hz. Following LiDAR4D (Zheng et al., 2024), we select 51 consecutive frames as a single scene and hold out 4 samples at 10-frame intervals for novel view synthesis (NVS) evaluation. For the nuScenes dataset, the LiDAR system uses 32 beams with a 40-degree vertical FOV and a 20Hz acquisition frequency. To ensure consistency with KITTI-360, we maintain a sampling frequency of 10Hz. We also provide the results on the Waymo (Sun et al., 2020) dataset in Appendix A.2.

**Competitors**   We evaluate our method alongside recent data-driven approaches LiDARsim (Manivasagam et al., 2020) and PCGen (Li et al., 2023). Additionally, we compare our results with the per-scene optimized reconstruction method NKSR (Huang et al., 2023), LiDAR-NeRF (Tao et al., 2023) and the state-of-the-art method, LiDAR4D (Zheng et al., 2024). We also include competitive dynamic neural radiance field methods, such as D-NeRF (Pumarola et al., 2021), K-Planes (Fridovich-Keil et al., 2023), and TiNeuVox (Fang et al., 2022), for comparison.

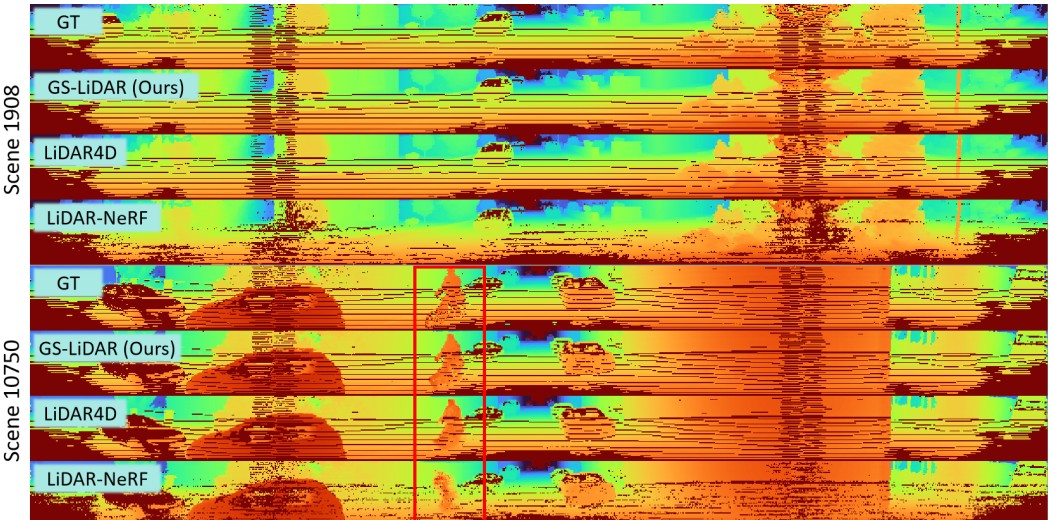

Figure 6: Comparison of the rendered depth map with competitors.

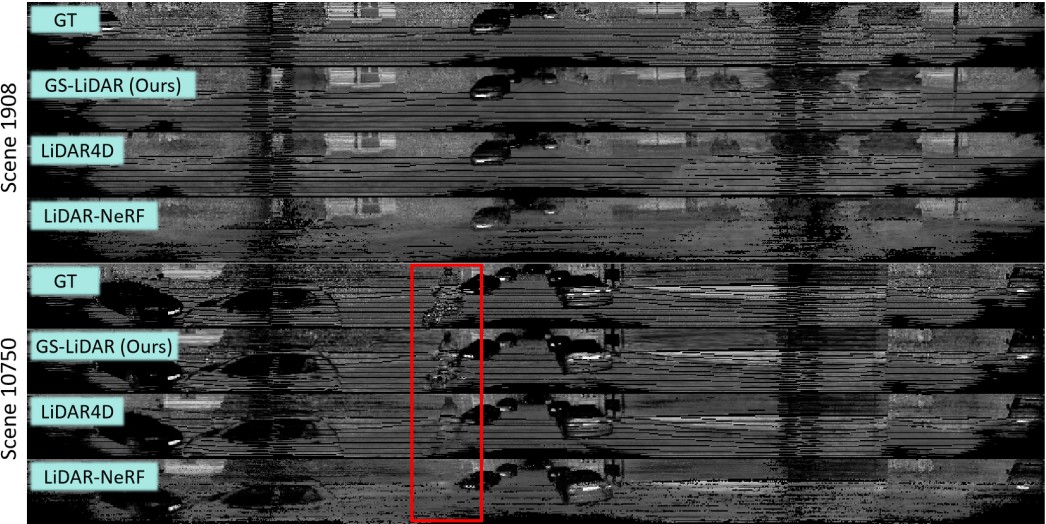

Figure 7: Comparison of the rendered intensity map with competitors.

**Metrics** We employ a comprehensive set of evaluation metrics for assessing point cloud, depth, and intensity measurements. Chamfer distance (Fan et al., 2017) is used to quantify the 3D geometric discrepancy between the generated and ground-truth point clouds based on nearest neighbors. Additionally, we report the F-score with a 5cm error threshold. For pixel-level error analysis of projected LiDAR range maps, we introduce Root Mean Square Error (RMSE) and Median Absolute Error (MedAE). Furthermore, we utilize LPIPS (Zhang et al., 2018), SSIM (Wang et al., 2004), and PSNR to assess image-level error on depth and intensity.

**Implementation details** We randomly sample $1 \times 10^6$ LiDAR points for point initialization. For the loss function and regularization terms, we use the following coefficients: $\lambda_d = 10$, $\lambda_{int} = 0.05$, $\lambda_{drop} = 0.05$, $\lambda_{dist} = 0.1$, $\lambda_n = 0.1$, and $\lambda_{ch} = 0.1$. All experiments are conducted on a single NVIDIA RTX A6000 GPU, with a total of 30,000 iterations, taking approximately 1.5 hours to produce the final results. The rendering speed reaches up to 11 frames per second (FPS).

## 4.2 EVALUATION ON STATIC SCENES

Table 1 provides the quantitative results for static scenes in KITTI-360 dataset across all methods. *GS-LiDAR* outperforms the competitors on most metrics. Notably, there is a 5.3% reduction in the Chamfer distance of the simulated point cloud, a 10.7% reduction in the RMSE of simulated

Table 4: **Ablation studies on various components of *GS-LiDAR*.**

| Method | Point Cloud | | Depth | | | | | Intensity | | | |
|---|---|---|---|---|---|---|---|---|---|---|---|
| | CD↓ | F-score↑ | RMSE↓ | MedAE↓ | LPIPS↓ | SSIM↑ | PSNR↑ | RMSE↓ | MedAE↓ | LPIPS↓ | SSIM↑ | PSNR↑ |
| **GS-LiDAR** | 0.1085 | 0.9231 | 3.1212 | 0.0340 | 0.0902 | 0.8553 | 28.4381 | 0.1161 | 0.0313 | 0.1825 | 0.5914 | 18.7482 |
| w/o ray-splat intersection | 3.1605 | 0.5091 | 6.6871 | 0.2367 | 0.4828 | 0.5754 | 21.5918 | 0.2320 | 0.1170 | 0.5166 | 0.1976 | 12.7069 |
| w/o periodic vibration | 0.1333 | 0.9100 | 3.1738 | 0.0359 | 0.0946 | 0.8577 | 28.2876 | 0.1162 | 0.0342 | 0.2166 | 0.5849 | 18.7288 |
| w/o median depth loss | 0.1254 | 0.9125 | 3.1414 | 0.0347 | 0.0941 | 0.8574 | 28.3567 | 0.1163 | 0.0313 | 0.1831 | 0.5902 | 18.7262 |
| w/o depth distortion loss | 0.1237 | 0.9229 | 3.2016 | 0.0341 | 0.0908 | 0.8567 | 28.2854 | 0.1176 | 0.0314 | 0.1842 | 0.5833 | 18.7078 |
| w/o normal consistency loss | 0.1158 | 0.9230 | 3.1993 | 0.0344 | 0.0908 | 0.8556 | 28.3852 | 0.1165 | 0.0314 | 0.1850 | 0.5887 | 18.7085 |
| w/o chamfer distance loss | 0.1152 | 0.9227 | 3.1354 | 0.0341 | 0.0907 | 0.8559 | 28.3892 | 0.1167 | 0.0341 | 0.1811 | 0.5911 | 18.7383 |
| w/o ray-drop refinement | 0.1121 | 0.9223 | 4.0791 | 0.0424 | 0.1952 | 0.7433 | 26.1083 | 0.1346 | 0.0384 | 0.2477 | 0.4569 | 17.4541 |

depth, and a 9.8% reduction in the RMSE of simulated intensity. As shown in Figure 5, *GS-LiDAR* produces a more cohesive LiDAR point cloud, which can be attributed to the accurate range maps generated by the proposed panoramic Gaussian splatting technique.

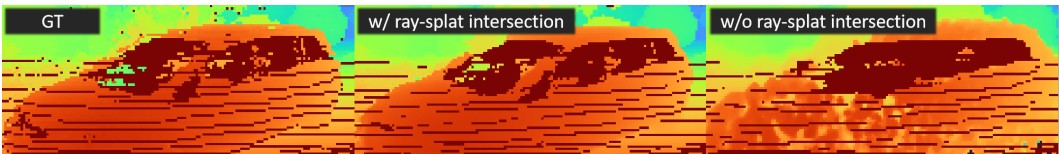

Figure 8: Comparison of w/ and w/o ray-splat intersection. A more detailed discussion is provided in Appendix A.1.

### 4.3 EVALUATION ON DYNAMIC SCENES

To further validate the effectiveness of *GS-LiDAR*, we conduct LiDAR synthesis evaluations on dynamic scenes from the KITTI-360 and nuScenes datasets. For the KITTI-360 dataset, as shown in Table 2, our method demonstrates superior performance, achieving a 0.3% reduction in chamfer distance for simulated point cloud, a 11.4% reduction in RMSE for simulated depth, and a 2.8% reduction in RMSE for simulated intensity. As illustrated in Figure 6 and Figure 7, *GS-LiDAR* achieves significantly better visual quality in simulated depth and intensity maps compared to competitors. This improvement is primarily due to the use of 2D Gaussian primitives with periodic vibration properties, enabling precise modeling of both static and dynamic geometries. For the nuScenes dataset, as shown in Table 3, *GS-LiDAR* also showcases notable performance, with a 2.5% reduction in chamfer distance for simulated point cloud, a 13.1% reduction in RMSE for simulated depth, and a 2.6% reduction in RMSE for simulated intensity.

### 4.4 ABLATION STUDY

We provide quantitative ablation studies on various components of *GS-LiDAR* in Table 4. As shown in Figure 8, we find that the use of ray-splat intersection enhances the capability of *GS-LiDAR* in modeling the surface of the scene, and a more detailed discussion is provided in Appendix A.1. For the "w/o ray-splat intersection" case, we implement a basic 3D Gaussian splatting approach by adapting the projection calculation method specifically for panoramic maps. The integration of periodic vibration properties further improves *GS-LiDAR*'s capability in handling dynamic elements. Regularization terms, including median depth loss, depth distortion loss, and chamfer distance loss, contribute to the improved quality of the simulated LiDAR point clouds. Additionally, the ray-drop refinement technique improves the accuracy of the ray-drop mask, resulting in substantial gains in the metrics for simulated depth and intensity.

## 5 CONCLUSION

We present *GS-LiDAR*, a novel framework designed to generate realistic LiDAR point clouds. To uniformly model the accurate surface of various elements in driving scenarios, we employ 2D Gaussian primitives with periodic vibration properties. Furthermore, we propose a novel panoramic Gaussian splatting technique with explicit ray-splat intersection for fast and efficient rendering of panoramic depth maps. By incorporating intensity and ray-drop SH coefficients into the Gaussian primitives, we enhance the realism of the rendered point clouds, making them more closely resemble actual LiDAR data. Our method significantly surpasses previous NeRF-based approaches in both computational speed and simulation quality on the KITTI-360 and nuScenes datasets.

ACKNOWLEDGMENT

This work was supported in part by National Natural Science Foundation of China (Grant No. 62376060).

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

# A APPENDIX

## A.1 RAY-SPLAT INTERSECTION

Although 3DGS performs well in camera image rendering, it is not suitable for panoramic LiDAR rendering. This limitation arises from the complexity of transforming 3D spatial points into LiDAR range map, which makes it challenging for the Jacobian matrix of the transformation to accurately approximate the effects of such a transformation. Given a point $(x, y, z)$ in the camera space, its corresponding radian angles $(\phi, \theta)$ can be calculated as:

$$\begin{pmatrix} \phi \\ \theta \end{pmatrix} = \begin{pmatrix} atan2(x, z) \\ atan2(\sqrt{x^2 + z^2}, -y) \end{pmatrix} \tag{22}$$

By combining this with Equation 8 from the paper, we can derive the projection of this point onto the LiDAR range map, resulting in its pixel coordinates $(\xi, \eta)$ as:

$$\begin{pmatrix} \xi \\ \eta \end{pmatrix} = \begin{pmatrix} \frac{W}{2\pi} atan2(x, z) + \frac{W}{2} \\ H \frac{atan2(\sqrt{x^2+z^2}, -y) - \text{VFOV}_{\min}}{(\text{VFOV}_{\max} - \text{VFOV}_{\min})} \end{pmatrix} \tag{23}$$

Thus, the Jacobian matrix can be computed as follows:

$$\mathrm{J} = \begin{pmatrix} \tilde{W} \frac{z}{x^2 + z^2} & 0 & -\tilde{W} \frac{x}{x^2 + z^2} \\ -\tilde{H} \frac{xy}{\sqrt{x^2+z^2}(x^2+y^2+z^2)} & \tilde{H} \frac{\sqrt{x^2+z^2}}{(x^2+y^2+z^2)} & -\tilde{H} \frac{yz}{\sqrt{x^2+z^2}(x^2+y^2+z^2)} \end{pmatrix} \tag{24}$$

where $\tilde{W} = \frac{W}{2\pi}$ and $\tilde{H} = \frac{H}{\text{VFOV}_{\max} - \text{VFOV}_{\min}}$.

We tile the plane with 2D and 3D Gaussian primitives of different colors and render their color and depth. As illustrated in Figure 9, we can observe that for 3D Gaussian primitives, due to the poor approximation of the Jacobian for the projection transformation, the originally tightly arranged points become scattered, which also leads to the discontinuity in depth shown in Figure 8 of the paper. In contrast, the 2D Gaussian primitives, being based on ray intersection without approximation, exhibit the desired effect.

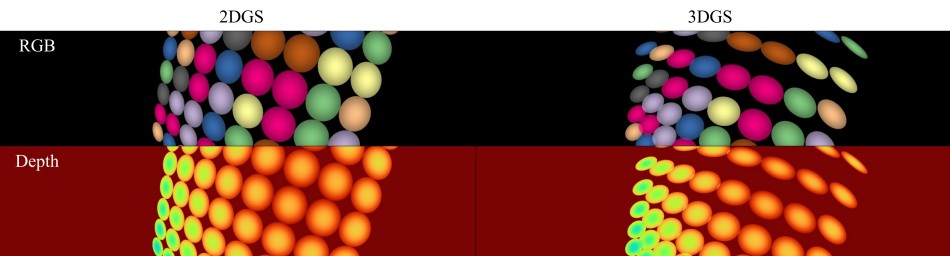

Figure 9: Ray-splat intersection v.s. 3D Gaussian splatting.

## A.2 EXPERIMENTS ON WAYMO DATASET

We additionally conducted experiments on sequences selected by PVG (Chen et al., 2023a) from the Waymo (Sun et al., 2020) dataset. Since methods like LiDAR4D (Zheng et al., 2024) have not been implemented on the Waymo dataset, we only report our own results in Table 5. The experimental results are consistent with those of KITTI-360 and nuScenes across the evaluation metrics, showing the scalability and generalization ability of our method. We have also additionally visualized LiDAR point clouds from the Waymo dataset in Figure 10 and Figure 11, showcasing the generalization ability of our approach again.

Table 5: **Metrics on Waymo dataset**.

| Method | Point Cloud | | Depth | | | | | Intensity | | | | |
|---|---|---|---|---|---|---|---|---|---|---|---|---|
| | CD↓ | F-score↑ | RMSE↓ | MedAE↓ | LPIPS↓ | SSIM↑ | PSNR↑ | RMSE↓ | MedAE↓ | LPIPS↓ | SSIM↑ | PSNR↑ |
| **GS-LiDAR (Ours)** | 0.2382 | 0.9055 | 5.8925 | 0.0198 | 0.0708 | 0.8394 | 22.2482 | 0.0415 | 0.0067 | 0.0627 | 0.7291 | 27.7420 |

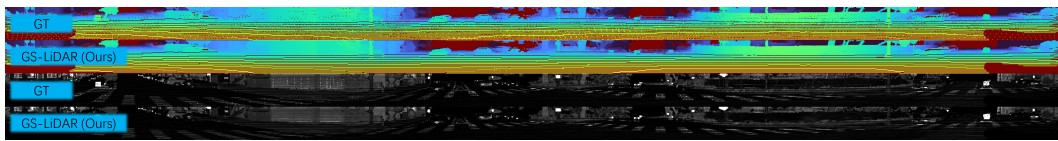

Figure 10: Comparison of the rendered depth and intensity map with ground truth. The first two rows depict the depth maps, while the subsequent two rows illustrate the intensity maps.

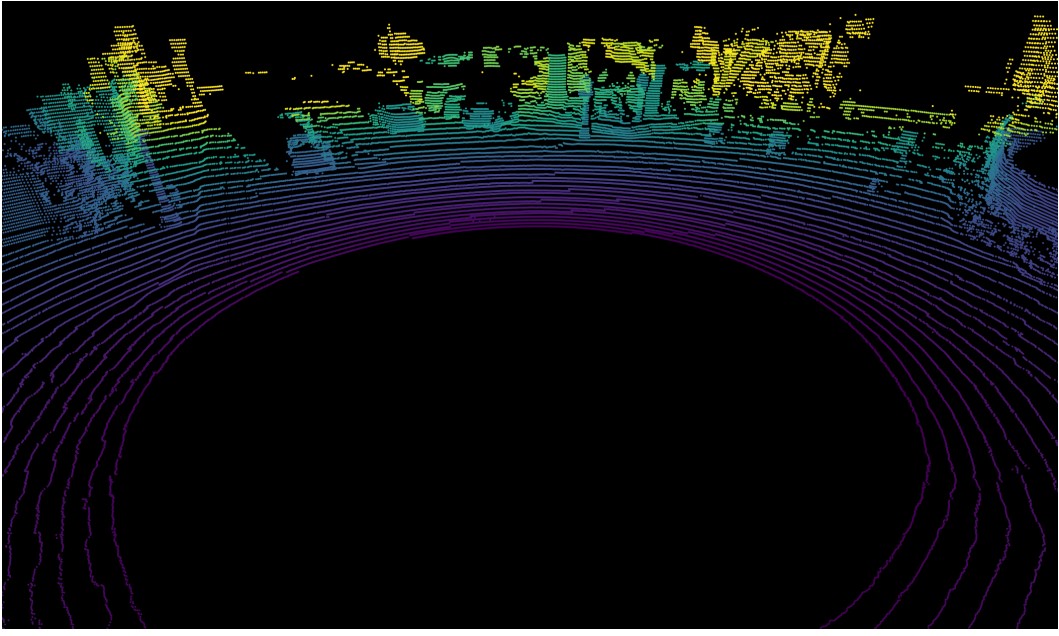

Figure 11: The visualization of LiDAR point clouds in three-dimensional space.

