# OpenReview forum: "GS-LiDAR: Generating Realistic LiDAR Point Clouds with Panoramic Gaussian Splatting"
_ICLR.cc/2025/Conference — ICLR 2025 Poster_

### Official Review · Reviewer_K7sa · 2024-10-24

**Soundness:** 3
**Presentation:** 2
**Contribution:** 3
**Rating:** 6
**Confidence:** 4

**Summary:**

In this paper, the authors present GS-LiDAR, a novel framework for generating realistic LiDAR point clouds. They employ 2D Gaussian primitives with periodic vibration properties to precisely model both static and dynamic elements in driving scenarios. Additionally, they introduce a panoramic rendering technique to obtain depth, intensity and ray-drop probability maps.

**Strengths:**

1)	The authors successfully take advantage of periodic vibration 2D Gaussian primitives to reconstruct dynamic point cloud from lidar measurements, and achieve state-of-the-art performance in terms of rendering speed and accuracy.

**Weaknesses:**

1）	Insufficient reference. The authors emphasized they employed 2D Gaussian primitives with periodic vibration properties and counted it as one of the contributions. However, periodic vibration and 2D Gaussian primitives are proposed by [1] and [2] respectively, which haven’t been credited in the introduction section.

[1] Huang, Binbin, et al. "2d gaussian splatting for geometrically accurate radiance fields." ACM SIGGRAPH 2024 Conference Papers. 2024.
[2] Yurui Chen, et al. "Periodic vibration gaussian: Dynamic urban scene reconstruction and real-time rendering.” arXiv preprint, 2023a.

2）	Ambiguous symbol. (u, v) in Line 204 and equation 2 denotes the target pixel of the rendering image, but later it becomes the coordinates within the local tangent plane space of 2D Gaussian in Line 245-255.

3）	Incorrect definition. The formulations of intensity map I in Line 316 and ray-drop probability map P_gs in Line 32 are identical.

**Questions:**

1）	The presentation of lidar rendering is not clear enough to make it difficult to follow. The authors highlight that they introduce a novel panoramic rendering technique with explicit ray-splat intersection, guided by panoramic LiDAR supervision. However, it is even unclear whether the image is rendered by ray marching, rasterizing or a hybrid approach.

---

> ### Author Response · Authors · 2024-11-23
>
> Thanks to the reviewer for the valuable feedback and constructive suggestions. We addressed the raised concerns in the revised manuscript and our response below.
>
> **Q1: Insufficient reference**
>
> Thanks for pointing this out!
> We have cited 2D Gaussian primitives and periodic vibration properties in the revised Introduction to clearly acknowledge their contributions and clarified our use of these concepts.
>
> **Q2: Ambiguous symbol**
>
> We apologize for the confusion.
> In Line 199 (previous 204) of the preliminary section, $\mathcal{G}^{\prime}$ represents the influence of the projected Gaussian primitive on the corresponding pixel in the rendering image.
> $\mathcal{G}(u, v)$ in Lines 240-250 (previous 245-255) refers to the influence of a 2D Gaussian within its local tangent space, where $(u, v)$ are the coordinates in that local tangent space.
> We have made these distinctions clearer in the revised manuscript.
>
> **Q3: Incorrect definition**
>
> Thanks. We compute the intensity map in Line 346 (previous 316) by aggregating the intensity values, $\lambda$, associated with each Gaussian. We compute the ray-drop probability map in Line 352 (previous 323), $P_\mathrm{gs}$, by aggregating the ray-drop probability values, $\rho$. We have clarified this distinction in the revised manuscript.
>
> **Q4: The presentation of lidar rendering is not clear enough to make it difficult to follow**
>
> Thanks. Our rendering approach is based on tile-based rasterization, similar to the method used in 3D Gaussian splatting (3DGS).
> While 3DGS evaluates the influence of a single primitive in 2D image space, it does not explicitly define the ray-splat intersection point, which can result in view inconsistencies. In contrast, our method leverages 2D Gaussian primitives, modeled as 2D disks in 3D space, allowing us to explicitly compute the ray-splat intersection point.
> By analytically deriving the ray-splat intersection during rasterization, our approach ensures view-consistent panoramic rendering.

---

> > ### Comment · Reviewer_K7sa · 2024-11-25
> >
> > Thank you for your rebuttal and the revised paper. I maintain my original score.

---

> > > ### Author Response · Authors · 2024-11-25
> > >
> > > We appreciate the reviewer's time for reviewing and thanks again for the valuable comments and the positive score!

---

> ### Author Response · Authors · 2024-11-25
>
> Dear Reviewer K7sa,
>
> We sincerely appreciate the reviewer's time for reviewing, and we really want to have a further discussion with the reviewer to see if our response solves the concerns. We have addressed all the thoughtful questions raised by the reviewer *(eg, insufficient reference,  ambiguous symbol and presentation of LiDAR rendering)* and we hope that our work’s impact and results are better highlighted with our responses. It would be great if the reviewer can kindly check our responses and provide feedback with further questions/concerns (if any). We would be more than happy to address them. Thank you!
>
> Best wishes,
>
> Authors

---

### Official Review · Reviewer_t1nR · 2024-11-03

**Soundness:** 3
**Presentation:** 3
**Contribution:** 3
**Rating:** 6
**Confidence:** 4

**Summary:**

This paper introduces a novel method for LiDAR novel view synthesis called GS-LiDAR. GS-LiDAR utilizes 2D Gaussian Splatting (2DGS) as a foundational representation for LiDAR reconstruction and incorporates periodic vibration for enhanced results. The reconstruction process is supervised by panoramic LiDAR data. The method is thoughtfully designed, incorporating various strategies to adapt 2DGS to LiDAR NVS scenarios, and is backed by extensive experimentation to demonstrate its performance. However, several concerns are noted below.

**Strengths:**

1. The paper is well-structured and easy to read.
2. The application of Gaussian Splatting to achieve LiDAR novel view synthesis is an innovative approach.
3. The method is well-conceived, integrating the unique properties of LiDAR point clouds into 2DGS.

**Weaknesses:**

1. The title of the paper suggests that the method is intended for “Generating” LiDAR point clouds. However, GS-LiDAR is designed for “reconstructing” LiDAR point cloud scenes. While the method does achieve novel view synthesis, the concept of reconstruction differs from generation. Generation [1][2] implies the creation of LiDAR point clouds from other user inputs, not just from an existing sequence of point clouds.
2. The rationale for using 2DGS instead of 3DGS as the primary representation is unclear. Although 2DGS has demonstrated superiority in geometric reconstruction, a comparative analysis between 2DGS and 3DGS (with the proposed strategies, rather than a standard 3DGS) would strengthen the paper.
3. The performance improvement over LiDAR4D, as shown in Tables 1 to 3, appears marginal. Additionally, the visualization results in Figure 4 do not clearly demonstrate significant performance gains. Including comparisons in video format would help substantiate the results.

[1] Ran, Haoxi, Vitor Guizilini, and Yue Wang. "Towards Realistic Scene Generation with LiDAR Diffusion Models." *Proceedings of the IEEE/CVF Conference on Computer Vision and Pattern Recognition*. 2024.

[2] Zyrianov, Vlas, et al. "LidarDM: Generative LiDAR Simulation in a Generated World." *arXiv preprint arXiv:2404.02903* (2024).

**Questions:**

1. Can the proposed method handle solid-state LiDAR?

---

> ### Author Response · Authors · 2024-11-23
>
> Thanks to the reviewer for the valuable feedback and constructive suggestions. We addressed the concerns in the revised manuscript and our response below.
>
> **Q1: ''Generating''**
>
> Yes, we are actually doing the novel view synthesis (NVS) task in this work.
> We use the term ''Generating'' as we aim to produce novel LiDAR point clouds from novel viewpoints.
> We are considering to change our title to ''Simulating Realistic Point Clouds with Panoramic Gaussian Splatting'' if it is allowed.
>
> **Q2: 2DGS instead of 3DGS**
>
> Great suggestion! Although 3DGS performs well in camera image rendering, it is not suitable for panoramic LiDAR rendering. This limitation arises from the complexity of transforming 3D spatial points into LiDAR range map, which makes it challenging for the Jacobian matrix of the transformation to accurately approximate the effects of such a transformation.
> Given a point $(x,y,z)$ in the camera space, its corresponding radian angles $(\phi, \theta)$ can be calculated as:
> $$
> \begin{pmatrix}
> \phi \\\\
> \theta
> \end{pmatrix}=
> \begin{pmatrix}
> atan2(x, z)\\\\
> atan2(\sqrt{x^2+z^2}, -y)
> \end{pmatrix}
> $$ By combining this with **Equation 8** from the paper, we can derive the projection of this point onto the LiDAR range map, resulting in its pixel coordinates $(\xi, \eta)$ as:
> $$
> \begin{pmatrix}
> \xi \\\\
> \eta
> \end{pmatrix}=
> \begin{pmatrix}
> \frac{W}{2\pi}atan2(x, z)+\frac{W}{2}\\\\
> H\frac{atan2(\sqrt{x^2+z^2}, -y) - \mathrm{VFOV}_{\min}}{ \mathrm{VFOV}\_{\max} - \mathrm{VFOV}\_{\min}}
> \end{pmatrix}
> $$ Thus, the Jacobian matrix can be computed as follows:
> $$
> \mathrm{J}=\begin{pmatrix}
> \tilde{W} \frac{z}{x^2 +z^2} & 0 & -\tilde{W} \frac{x}{x^2 +z^2} \\\\
> -\tilde{H} \frac{xy}{\sqrt{x^2+z^2}(x^2+y^2+z^2)} & \tilde{H}\frac{\sqrt{x^2+z^2}}{(x^2+y^2+z^2)} & -\tilde{H} \frac{yz}{\sqrt{x^2+z^2}(x^2+y^2+z^2)}
> \end{pmatrix}
> $$ where $\tilde{W} = \frac{W}{2\pi}$ and $\tilde{H} = \frac{H}{\mathrm{VFOV}\_{\max}-\mathrm{VFOV}\_{\min}}$.
>
> We tile the plane with 2D and 3D Gaussian primitives of different colors and render their color and depth. Please refer to the supplementary material ```2dgs_vs_3dgs.pdf```. We can observe that for 3D Gaussian primitives, due to the poor approximation of the Jacobian for the projection transformation, the originally tightly arranged points become scattered, which also leads to the discontinuity in depth shown in **Figure 8** of the updated paper. In contrast, the 2D Gaussian primitives, being based on ray intersection without approximation, exhibit the desired effect.
>
> **Q3: Performance**
>
> In the field of autonomous driving, the most critical factor for LiDAR simulation is rendering speed.  Efficient rendering is essential to meet the requirements of closed-loop training and evaluation for driving agents.  With efficient rendering, driving agents can undergo closed-loop training on huge datasets, bridging the final gap between simulation and real-world driving.  Compared to LiDAR4D's 0.3 fps, GS-LiDAR rendering speed reaches 11 fps, which is 37 times faster, marking a key advancement in the usability of LiDAR simulation.  This improvement in rendering speed is attributed to the carefully designed ray-splat intersection technique for LiDAR range image rendering and the efficient CUDA implementation.
>
> Furthermore, in comparison with LiDAR4D, our method shows improvements across various metrics, demonstrating the reliability of our approach. For static scenes on the KITTI-360 dataset, GS-LiDAR achieves a 5.3\% reduction in Chamfer distance for the simulated point cloud, a 10.7\% reduction in RMSE for simulated depth, and a 9.8\% reduction in RMSE for simulated intensity compared to the leading LiDAR4D. For dynamic scenes on KITTI-360 and nuScenes, GS-LiDAR achieves 11.4\% and 13.1\% reduction in RMSE for simulated depth, respectively.
>
> Additionally, existing mainstream method for LiDAR processing by driving agents is voxel occupancy representation.  As shown in **Figure 5** (previous Figure 4), by leveraging supervision on median depth, our method is able to better model the edges of objects without scattering, which is a significant improvement for obtaining realistic and reliable voxel occupancy, an advancement that might be overlooked as it is not directly reflected in the metrics.
>
> **Q4: Can the proposed method handle solid-state LiDAR?**
>
> Yes, our method is fully capable of handling solid-state LiDAR, as it shares similar properties with conventional mechanical LiDAR, such as intensity and ray-drop. Solid-state LiDAR can be treated in the same way as mechanical LiDAR within our framework.

---

> > ### Comment · Reviewer_t1nR · 2024-11-25
> > **Response to Rebuttal.**
> >
> > The rebuttal has successfully addressed most of my concerns regarding the title and the choice of 2DGS. While I still have some concerns about the technical contribution (similar to those raised by reviewers aQ3K and YP8o), I remain positive overall, as this is the first paper to apply GS to LiDAR NVS.
> > I will maintain my original score.

---

> ### Author Response · Authors · 2024-11-25
>
> Dear Reviewer t1nR
>
> We sincerely appreciate the reviewer's time for reviewing and thanks again for the valuable comments and the positive score!
>
> Regarding the technical contribution, we strongly advocat that using Gaussian splatting for LiDAR  simulation is **non-trival**. Simply applying 3D Gaussian primitives for point cloud reconstruction revealed that the nonlinear transformation from camera-space coordinates to range image pixels is highly complex. **A naive approximation using the Jacobian matrix as a linear transformation is inadequate**, leading to geometric inconsistencies across multiple viewpoints.
>
> To this end, we propose a novel **panoramic Gaussian splatting** technique. This involves parameterizing the rays corresponding to panoramic range image pixels and intersecting them with 2DGS. **We implemented this process efficiently using CUDA**, making it feasible to model LiDAR using Gaussian primitives and perform efficient rendering.
>
> Compared to the previous NeRF-based LiDAR4D approach, our method not only achieves improvements in metrics but also demonstrates a 37× increase in rendering speed. This significant advancement is attributed to our advanced panoramic rendering technique and its highly efficient implementation. This represents a significant step toward LiDAR simulation for autonomous driving. In a nutshell, we believe our work lays the foundation for LiDAR simulation in autonomous driving.
>
> Hope our responses clarify above thoughtful question. It is very much appreciated if the reviewer can kindly check our response, and *could* increase the score of our paper and champion it for publication. Thank you!
>
> Best wishes
>
> Authors

---

### Official Review · Reviewer_YP8o · 2024-11-03

**Soundness:** 3
**Presentation:** 3
**Contribution:** 2
**Rating:** 5
**Confidence:** 3

**Summary:**

This paper addresses the task of LIDAR novel view synthesis and introduces a framework called GS-LIDAR, which generates realistic LiDAR point clouds using panoramic Gaussian splatting. To accurately reconstruct both static and dynamic elements in driving scenarios, the authors utilize 2D Gaussian primitives with periodic vibration properties. Additionally, the paper presents a novel panoramic Gaussian splatting technique that incorporates explicit ray-splat intersection for rapid and efficient rendering of panoramic depth maps. Experimental results demonstrate the framework's state-of-the-art performance in terms of quantitative metrics, visual quality, and efficiency.

**Strengths:**

The paper is generally well-presented, with a system architecture that is clearly outlined and easy to follow. Additionally, the experimental results—encompassing both quantitative and qualitative evaluations—are impressive, demonstrating performance that surpasses the current state-of-the-art in this specific task. Furthermore, the ablation study is thorough and effectively substantiates the proposed approach.

**Weaknesses:**

1) This article appears to present an incremental improvement rather than a significant innovation. The primary contribution seems to be the replacement of NeRF in LIDAR4D with Gaussian Splatting. While this method offers advantages in training and rendering speed compared to LIDAR4D, these benefits are inherently tied to the properties of Gaussian Splatting itself.
2) Regarding the methodology and experiments, the most notable innovation appears to be the Ray-splat intersection in panoramic Gaussian Splatting. However, this aspect is not clearly articulated in the methods section. For instance, in Line 249, the physical significance of the variables u and v is not intuitively conveyed, nor is the derivation of the transformation from UV space to screen space adequately explained. The authors should provide a more detailed explanation of this section or include a schematic diagram  (i.e. a geometric illustration) to enhance reader comprehension.
3) Furthermore, the authors need to conduct a thorough analysis of how the Ray-splat intersection, as demonstrated in the ablation experiment, enhances the capability of GS-LiDAR in accurately modeling surface features of the scene and leads to significant performance improvements.

**Questions:**

1. The novelty of the proposed method, especially in comparison to LIDAR4D, requires clearer articulation, which should focus on  how the paper's contributions go beyond simply replacing NeRF with Gaussian Splatting.
2. The Method section includes extensive background information on existing techniques, such as ray-drop refinement and various loss functions, which do not directly contribute to the core innovations of the paper. Consider condensing these parts or moving them to the supplementary materials. What's more, a more detailed explanation and accompanying diagram (i.e. a geometric illustration) of the Ray-splat intersection would improve clarity and focus.
3. Please explain in detail how the addition of the Ray-splat intersection mechanism significantly enhances the performance of the proposed method.
4. Additional experiments and analyses, such as testing on the Waymo dataset  with different sensor configurations or environmental conditions, would further validate the effectiveness and generalization ability of the approach.
5. Typo: In Line 323, the ray-drop probability signal $\rho_{k}$ is wrongly spelled by $\lambda_{k}$.

---

> ### Author Response · Authors · 2024-11-23
>
> Thanks to the reviewer for the valuable feedback and constructive suggestions. We addressed the raised concerns in the revised manuscript and our response below.
>
> **Q1: Novelty**
>
> Closed-loop simulation for autonomous driving urgently requires efficient rendering of LiDAR point clouds from novel viewpoints to facilitate multimodal driving agent training and evaluation. However, existing LiDAR reconstruction and novel view synthesis methods are predominantly based on NeRF, whose training and inference speeds are prohibitively slow, severely limiting progress in fields such as autonomous driving.
>
> Motivated by this limitation, using 3D Gaussian primitives for point cloud reconstruction naturally emerges as a promising idea. However, our experiments revealed that the nonlinear transformation from camera-space coordinates to range image pixels is highly complex. A naive approximation using the Jacobian matrix as a linear transformation is inadequate, leading to geometric inconsistencies across multiple viewpoints.
>
> To address this issue, we shifted our focus to 2D Gaussian primitives, which avoid approximations by leveraging ray-splat intersections. To enable 2DGS for LiDAR range image rendering, we proposed a novel panoramic Gaussian splatting technique. This involves parameterizing the rays corresponding to panoramic range image pixels and intersecting them with 2DGS. We implemented this process efficiently using CUDA, making it feasible to model LiDAR using Gaussian primitives and perform efficient rendering. To this end, utilizing 2D Gaussian as the representation for LiDAR simulation is non-trival.
>
> Compared to the previous NeRF-based LiDAR4D approach, our method not only achieves improvements in metrics but also demonstrates a 37× increase in rendering speed. This significant advancement is attributed to our advanced panoramic rendering technique and its highly efficient implementation. This represents a significant step toward LiDAR simulation for autonomous driving.
>
> Furthermore, through extensive experiments, we enhanced each Gaussian primitive with additional attributes, including intensity and ray-drop probability. We also incorporated periodic viberation property, enabling the Gaussian primitives to model dynamic scenarios. **In a nutshell, we believe our work *per se* is novel and our contribution lays the foundation for LiDAR simulation in autonomous driving.**
>
> **Q2: Ray-splat intersection**
>
> Thanks. $(u, v)$ are the coordinates on the local tangent plane of the 2D Gaussian primitive, with the coordinate axes aligned with the directions $t_u$ and $t_v$ in the world coordinate system. To transform this coordinate into the world coordinate system, the following steps are performed:
>
> (i) Apply scaling by $s_u$ and $s_v$ along the corresponding axes, resulting in the relative position with respect to the 2DGS center: $s_u t_u u + s_v t_v v$.
>
> (ii) Translate this relative position by the 2DGS center $\tilde{\mu}$, yielding the world coordinates: $\tilde{\mu} + s_u t_u u + s_v t_v v$
>
> To transform the coordinates into the camera coordinate system, the world-to-camera transformation matrix $𝑊$ should be applied. The final transformation can be expressed as: $W(\tilde{\mu} + s_u t_u u + s_v t_v)$, which is consistent with **Equation 6** in the paper.
>
> We have also included an illustration in the updated paper; please refer to **Figure 4**.

---

> > ### Author Response · Authors · 2024-11-23
> >
> > **Q3: Ablation of ray-splat intersection**
> >
> > Although 3DGS performs well in camera image rendering, it is not suitable for panoramic LiDAR rendering. This limitation arises from the complexity of transforming 3D spatial points into LiDAR range map, which makes it challenging for the Jacobian matrix of the transformation to accurately approximate the effects of such a transformation.
> > Given a point $(x,y,z)$ in the camera space, its corresponding radian angles $(\phi, \theta)$ can be calculated as:
> > $$
> > \begin{pmatrix}
> > \phi \\\\
> > \theta
> > \end{pmatrix}=
> > \begin{pmatrix}
> > atan2(x, z)\\\\
> > atan2(\sqrt{x^2+z^2}, -y)
> > \end{pmatrix}
> > $$ By combining this with **Equation 8** from the paper, we can derive the projection of this point onto the LiDAR range map, resulting in its pixel coordinates $(\xi, \eta)$ as:
> > $$
> > \begin{pmatrix}
> > \xi \\\\
> > \eta
> > \end{pmatrix}=
> > \begin{pmatrix}
> > \frac{W}{2\pi}atan2(x, z)+\frac{W}{2}\\\\
> > H\frac{atan2(\sqrt{x^2+z^2}, -y) - \mathrm{VFOV}_{\min}}{(\mathrm{VFOV}\_{\max}-\mathrm{VFOV}\_{\min})}
> > \end{pmatrix}
> > $$ Thus, the Jacobian matrix can be computed as follows:
> > $$
> > \mathrm{J}=\begin{pmatrix}
> > \tilde{W} \frac{z}{x^2 +z^2} & 0 & -\tilde{W} \frac{x}{x^2 +z^2} \\\\
> > -\tilde{H} \frac{xy}{\sqrt{x^2+z^2}(x^2+y^2+z^2)} & \tilde{H}\frac{\sqrt{x^2+z^2}}{(x^2+y^2+z^2)} & -\tilde{H} \frac{yz}{\sqrt{x^2+z^2}(x^2+y^2+z^2)}
> > \end{pmatrix}
> > $$ where $\tilde{W} = \frac{W}{2\pi}$ and $\tilde{H} = \frac{H}{\mathrm{VFOV}\_{\max}-\mathrm{VFOV}\_{\min}}$.
> >
> > We tile the plane with 2D and 3D Gaussian primitives of different colors and render their color and depth. Please refer to the supplementary material ```2dgs_vs_3dgs.pdf```. We can observe that for 3D Gaussian primitives, due to the poor approximation of the Jacobian for the projection transformation, the originally tightly arranged points become scattered, which also leads to the discontinuity in depth shown in **Figure 8** of the updated paper. In contrast, the 2D Gaussian primitives, being based on ray intersection without approximation, exhibit the desired effect.
> >
> > **Q4: Experiments on Waymo dataset**
> >
> > We additionally conducted experiments on sequences selected by PVG from the Waymo dataset.
> > Since methods like LiDAR4D have not been implemented on the Waymo dataset, we only report our own results in the table below.
> > The experimental results are consistent with those of KITTI-360 and nuScenes across the evaluation metrics, showing the scalability and generalization ability of our method.
> >
> >
> > |          | point cloud |         | depth  |        |        |        |         | intensity |        |        |        |         |
> > | -------- | ----------- | ------- | ------ | ------ | ------ | ------ | ------- | --------- | ------ | ------ | ------ | ------- |
> > |          | CD          | F-score | RMSE   | MedAE  | LPIPS  | SSIM   | PSNR    | RMSE      | MedAE  | LPIPS  | SSIM   | PSNR    |
> > | GS-LiDAR | 0.1738      | 0.9020  | 6.1541 | 0.0317 | 0.1797 | 0.7213 | 22.5714 | 0.0825    | 0.0103 | 0.2176 | 0.6568 | 21.7154 |
> >
> > We have also additionally visualized consecutive frame LiDAR point clouds from the Waymo dataset in the supplementary materials ```waymo.mp4```, showcasing the generalization ability of our approach again.
> >
> > **Q5: Typo: ray-drop probability**
> >
> > Yes, in Line 352 (previous 323), we compute the ray-drop probability map $P_\mathrm{gs}$ by aggregating the ray-drop probability values $\rho_k$. We have clarified this in Line 352 (previous 323) of the revised manuscript.

---

> ### Author Response · Authors · 2024-11-25
>
> Dear Reviewer YP8o,
>
> We sincerely appreciate the reviewer's time for reviewing, and we really want to have a further discussion with the reviewer to see if our response solves the concerns. We have addressed all the thoughtful questions raised by the reviewer *(eg, novelty,  ablation of ray-splat intersection and experiments on Waymo dataset)* and we hope that our work’s impact and results are better highlighted with our responses. It would be great if the reviewer can kindly check our responses and provide feedback with further questions/concerns (if any). We would be more than happy to address them. Thank you!
>
> Best wishes,
>
> Authors

---

> > ### Author Response · Authors · 2024-11-26
> >
> > Dear Reviewer YP8o,
> >
> > We sincerely thank the reviewer for the valuable comments and suggestions. As the discussion phase is nearing its end, we wondered if the reviewer might still have any concerns that we could address. We believe our responses on *novelty, illustration and ablation of ray-splat intersection, experiments on Waymo dataset* addressed all the questions/concerns, and hope our response helps the final recommendation. Thank you!
> >
> > Best wishes,
> >
> > Authors

---

> > > ### Comment · Reviewer_YP8o · 2024-11-27
> > >
> > > Thanks for the author's efforts during the rebuttal. My concerns about experiments and illustration are mostly tackeld. I would increase my score this point, but I still think the novelty of this paper is below the average bar of ICLR. Just as the comments from Reviewer aQ3K , it is 'limited to a combination of existing methods'.

---

> ### Author Response · Authors · 2024-11-27
>
> Dear Reviewer YP8o,
>
> We genuinely appreciate the reviewer’s recognition of our rebuttal, as well as reviewer's valuable time and efforts for reviewing our work.
>
> Honestly speaking, the quality of a piece of work should not be judged solely on how new is the method, but more on **what problem it solved and to what extend it solved the problem**.
>
> In our work, we are the first to introduce Gaussian splatting into LiDAR NVS, utilizing panoramic ray-splat intersection in LiDAR range image rendering and addressing the geometric inconsistencies across multiple viewpoints that arise from the naive use of 3D Gaussians. Our approach successfully handles static and dynamic scenes, while enabling fast rendering capabilities.
>
> **Neither the existing techniques, nor *a combination of existing methods*, could solve this problem**. The concept of 3DGS first appeared on arXiv on August 8, 2023, while the most recent LiDAR NVS method, LiDAR4D (CVPR, March 6, 2024), still relies on NeRF. **This highlights the non-triviality and challenges of proposing GS-based method into the LiDAR NVS**. In other words, we firmly believe our approach *per se* is novel, and hope our panoramic rendering method could set a new paradigm for future GS-based LiDAR NVS.
>
>
> As Reviewer aQ3K noted, "This paper still makes significant contributions to LiDAR NVS."
>
> Hope our responses clarify on the point of “novelty”. It is very much appreciated if the reviewer could champion our work for publication. Thank you!
>
> Best wishes,
>
> Authors

---

### Official Review · Reviewer_aQ3K · 2024-11-06

**Soundness:** 3
**Presentation:** 3
**Contribution:** 2
**Rating:** 6
**Confidence:** 4

**Summary:**

This paper proposes GS-LiDAR, which introduces a novel framework for LiDAR novel view synthesis (NVS). Compared to previous NeRF-based works designed with fixed resolution and symmetrical structure, this paper employs 2D Gaussian primitives with periodic vibration properties to reconstruct the dynamic driving scenarios more effectively. Furthermore, based on the panoramic LiDAR rendering technique, the authors calculate the ray-splat intersection to achieve explicit and accurate results. And the spherical harmonic (SH) of Gaussian primitives can also support intensity and ray-drop reconstruction. Experiments on KITTI-360 and nuScenes demonstrate the superiority of GS-LiDAR in reconstruction quality and rendering efficiency.

**Strengths:**

- Previous NeRF-based works with Hashgrid have a symmetrical structure and fixed resolution for each axis, which is not suitable and effective enough for driving scenarios. This paper first adopts GS representation for LiDAR NVS, making the reconstruction and rendering more efficient.

- 2D Gaussian primitives have higher geometric accuracy. Moreover, it incorporates the periodic vibration properties, allowing for both static and dynamic scene reconstruction.

- Based on the projection nature of LiDAR point clouds, this paper utilizes the panoramic rendering technique to accommodate Gaussian Splatting, and further proposes ray-splat intersection to obtain more accurate rendering results.

**Weaknesses:**

- In terms of the core methodology, this paper is more of a combination of existing methods (2D-GS and PVG) and does not substantially innovate them, which results in a limited contribution. It should include more insights and analysis of the methods and why they are well suited for LiDAR-NVS.

- The concept of panoramic Gaussian splatting is a little bit confusing in Section 3.3. And according to the proposed ray-splat intersection, it's more like the "ray tracing" manner than the "splatting" one. If so, it needs to compute the intersection of every ray and all Gaussian primitives, which is still not efficient enough.

- Despite the efficiency gains due to the Gaussian itself, the final rendering results don't seem to be a huge improvement.

**Questions:**

Besides the concerns mentioned in the weaknesses, there are additional questions as follows:

- It introduces normal consistency term in the loss function and uses depth gradient as pseudo ground-truth. However, will it produce incorrect results due to LiDAR projection, ray-drop or at object edges? Is it more plausible to use 3D point clouds for normal estimation? In addition, a comparison of this loss term is missing in the ablation study.

- How to initialize Gaussian points for a dynamic scene? If all point clouds are aggregated and sampled, this may lead to unreasonable initialization of dynamic objects (especially with long trajectories). Furthermore, are different scenes with different sequence lengths and ranges initialized using the same number of points? And how about the final number of Gaussian points?

- As can be seen in Figure 1, the results of 3D-GS are very poor, but the paper lacks details of its implementation and the analysis of its failure. Since 3D-GS could not be directly migrated to LiDAR-NVS, relevant explanations are necessary.

- Is it reasonable to model long trajectory motion with periodic vibration properties? It would be better to visualize the motion trajectories of Gaussian primitives to prove their temporal consistency.

---

> ### Author Response · Authors · 2024-11-23
>
> Thanks for the valuable feedback and constructive suggestions. We addressed the raised concerns in the revised manuscript and our response below.
>
> **Q1: Contribution**
>
> Closed-loop simulation for autonomous driving urgently requires efficient rendering of LiDAR point clouds from novel viewpoints to facilitate multimodal driving agent training and evaluation. However, existing LiDAR reconstruction and novel view synthesis methods are predominantly based on NeRF, whose training and inference speeds are prohibitively slow, severely limiting progress in fields such as autonomous driving.
>
> Motivated by this limitation, using 3D Gaussian primitives for point cloud reconstruction naturally emerges as a promising idea. However, our experiments revealed that the nonlinear transformation from camera-space coordinates to range image pixels is highly complex. A naive approximation using the Jacobian matrix as a linear transformation is inadequate, leading to geometric inconsistencies across multiple viewpoints.
>
> To address this issue, we shifted our focus to 2D Gaussian primitives, which avoid approximations by leveraging ray-splat intersections. To enable 2DGS for LiDAR range image rendering, we proposed a novel panoramic Gaussian splatting technique. This involves parameterizing the rays corresponding to panoramic range image pixels and intersecting them with 2DGS. We implemented this process efficiently using CUDA, making it feasible to model LiDAR using Gaussian primitives and perform efficient rendering. To this end, utilizing 2D Gaussian as the representation for LiDAR simulation is non-trival.
>
> Compared to the previous NeRF-based LiDAR4D approach, our method not only achieves improvements in metrics but also demonstrates a 37× increase in rendering speed. This significant advancement is attributed to our advanced panoramic rendering technique and its highly efficient implementation. This represents a significant step toward LiDAR simulation for autonomous driving.
>
> Furthermore, through extensive experiments, we enhanced each Gaussian primitive with additional attributes, including intensity and ray-drop probability. We also incorporated periodic viberation property, enabling the Gaussian primitives to model dynamic scenarios. **In a nutshell, we believe our work *per se* is novel and our contribution lays the foundation for LiDAR simulation in autonomous driving.**
>
> **Q2: Ray-splat intersection**
>
> Although we calculate the intersection of each ray with the Gaussian primitives, we still adopt the tile-based rendering method proposed in 3D Gaussian splatting to achieve efficient rendering.
>
> As illustrated in **Figure 4** in the updated paper, we first transform the epipolar coordinates into pixel coordinates and sort the Gaussian primitives within each tile. For pixel rendering, the $\alpha$ and depth are determined based on the intersection results.
>
>
> **Q3: Performance improvement**
>
> In the field of autonomous driving, the most critical factor for LiDAR simulation is rendering speed. Efficient rendering is essential to meet the requirements of closed-loop training and evaluation for driving agents. With efficient rendering, driving agents can undergo closed-loop training on huge datasets, bridging the final gap between simulation and real-world driving. Compared to LiDAR4D's 0.3 fps, GS-LiDAR rendering speed reaches 11 fps, which is 37 times faster, marking a key advancement in the usability of LiDAR simulation.
> This improvement in rendering speed is attributed to the carefully designed ray-splat intersection technique for LiDAR range image rendering and the efficient CUDA implementation.
>
> Furthermore, in comparison with LiDAR4D, our method shows improvements across various metrics, demonstrating the reliability of our approach. For static scenes on the KITTI-360 dataset, GS-LiDAR achieves a 5.3\% reduction in Chamfer distance for the simulated point cloud, a 10.7\% reduction in RMSE for simulated depth, and a 9.8\% reduction in RMSE for simulated intensity compared to the leading LiDAR4D. For dynamic scenes on KITTI-360 and nuScenes, GS-LiDAR achieves 11.4\% and 13.1\% reduction in RMSE for simulated depth, respectively.
>
> Additionally, existing mainstream method for LiDAR processing by driving agents is voxel occupancy representation.
> As shown in **Figure 5** (previous Figure 4), by leveraging supervision on median depth, our method is able to better model the edges of objects without scattering, which is a significant improvement for obtaining realistic and reliable voxel occupancy, an advancement that might be overlooked as it is not directly reflected in the metrics.

---

> ### Author Response · Authors · 2024-11-23
>
> **Q4: Normal consistency**
>
> For the rendered depth map, we first back-project it into 3D space to obtain a 3D point cloud. The spatial gradients of the 3D point cloud are then used as pseudo ground-truth normals. Since the depth map and ray-drop are rendered separately, the resulting depth map is dense and unaffected by ray-drop. While object edges may affect the accuracy of this loss, their sparsity and the relatively low weight assigned to this loss ensure that it optimizes the geometric consistency of surfaces without disrupting the overall structure of the scene.
> We additionally add the ablation experiment of normal consistency loss below and in **Table 4** of the updated paper.
> |          | point cloud |         | depth  |        |        |        |         | intensity |        |        |        |         |
> | -------- | ----------- | ------- | ------ | ------ | ------ | ------ | ------- | --------- | ------ | ------ | ------ | ------- |
> |          | CD$\downarrow$         | F-score$\uparrow$ | RMSE$\downarrow$   | MedAE$\downarrow$  | LPIPS$\downarrow$  | SSIM$\uparrow$   | PSNR$\uparrow$    | RMSE$\downarrow$      | MedAE$\downarrow$  | LPIPS$\downarrow$  | SSIM$\uparrow$   | PSNR$\uparrow$    |
> | GS-LiDAR | 0.1085 | 0.9231| 3.1212 |0.0340 |0.0902 |0.8553 |28.4381 |0.1161 |0.0313 |0.1825 |0.5914 |18.7482 |
> | w/o normal consistency loss | 0.1158 | 0.9230 | 3.1993 |0.0344 |0.0908 |0.8556 |28.3852 |0.1165 |0.0314 |0.1850 |0.5887 |18.7085 |
>
> We can observe that the normal consistency loss, by enhancing the geometric consistency in 3D space, significantly reduces the chamfer distance (CD) between the rendered point cloud and the ground truth point cloud in 3D space.
>
> **Q5: Initialize Gaussian points**
>
> For dynamic scenes, we initialize the positions of Gaussians by sampling from the LiDAR point clouds of each view. The life peak property $\tau$ of each Gaussian is initialized based on the timestamp of the view from which it was sampled. As our Gaussian primitives incorporate time-dependent opacity (Eq. 4), which peaks at the life peak $\tau$ and decays over time, this ensures a reasonable initialization where Gaussians sampled at a given timestamp primarily influence adjacent time periods. We initialize all scenes with $1 \times 10^6$ Gaussians. After optimization, the final number of Gaussians typically ranges from approximately $5 \times 10^6$ to $1 \times 10^7$, depending on the scene's complexity.
>
> **Q6: 3DGS in Figure 1**
>
> Great suggestion!
> Although 3DGS performs well in camera image rendering, it is not suitable for panoramic LiDAR rendering. This limitation arises from the complexity of transforming 3D spatial points into LiDAR range map, which makes it challenging for the Jacobian matrix of the transformation to accurately approximate the effects of such a transformation.
> Given a point $(x,y,z)$ in the camera space, its corresponding radian angles $(\phi, \theta)$ can be calculated as:
> $$
> \begin{pmatrix}
> \phi \\\\
> \theta
> \end{pmatrix}=
> \begin{pmatrix}
> atan2(x, z)\\\\
> atan2(\sqrt{x^2+z^2}, -y)
> \end{pmatrix}
> $$ By combining this with **Equation 8** from the paper, we can derive the projection of this point onto the LiDAR range map, resulting in its pixel coordinates $(\xi, \eta)$ as:
> $$
> \begin{pmatrix}
> \xi \\\\
> \eta
> \end{pmatrix}=
> \begin{pmatrix}
> \frac{W}{2\pi}atan2(x, z)+\frac{W}{2}\\\\
> H\frac{atan2(\sqrt{x^2+z^2}, -y) - \mathrm{VFOV}_{\min}}{(\mathrm{VFOV}\_{\max}-\mathrm{VFOV}\_{\min})}
> \end{pmatrix}
> $$ Thus, the Jacobian matrix can be computed as follows:
> $$
> \mathrm{J}=\begin{pmatrix}
> \tilde{W} \frac{z}{x^2 +z^2} & 0 & -\tilde{W} \frac{x}{x^2 +z^2} \\\\
> -\tilde{H} \frac{xy}{\sqrt{x^2+z^2}(x^2+y^2+z^2)} & \tilde{H}\frac{\\sqrt{x^2+z^2}}{(x^2+y^2+z^2)} & -\tilde{H} \frac{yz}{\\sqrt{x^2+z^2}(x^2+y^2+z^2)}
> \end{pmatrix}
> $$ where $\tilde{W} = \frac{W}{2\pi}$ and $\tilde{H} = \frac{H}{\mathrm{VFOV}\_{\max}-\mathrm{VFOV}\_{\min}}$.
>
> We tile the plane with 2D and 3D Gaussian primitives of different colors and render their color and depth. Please refer to the supplementary material ```2dgs_vs_3dgs.pdf```. We can observe that for 3D Gaussian primitives, due to the poor approximation of the Jacobian for the projection transformation, the originally tightly arranged points become scattered, which also leads to the discontinuity in depth shown in **Figure 8** of the updated paper. In contrast, the 2D Gaussian primitives, being based on ray intersection without approximation, exhibit the desired effect.
>
> **Q7: Motion trajectories**
>
> Our method is able to render the velocity of the Gaussian points. We additionally provided the visualization video ```pvg_trajectory.mp4``` in the supplementary materials. As shown in the video, moving objects in the scene have higher velocities (blue), while static objects exhibit velocities close to zero (red). This demonstrates that multiple high-speed, short-period periodic vibration Gaussian points can collectively represent long trajectory motion effectively.

---

> ### Author Response · Authors · 2024-11-25
>
> Dear Reviewer aQ3K,
>
> We sincerely appreciate the reviewer's time for reviewing, and we really want to have a further discussion with the reviewer to see if our response solves the concerns. We have addressed all the thoughtful questions raised by the reviewer *(eg, contribution, normal consistency and 2DGS v.s. 3DGS)* and we hope that our work’s impact and results are better highlighted with our responses. It would be great if the reviewer can kindly check our responses and provide feedback with further questions/concerns (if any). We would be more than happy to address them. Thank you!
>
> Best wishes,
>
> Authors

---

> > ### Comment · Reviewer_aQ3K · 2024-11-25
> >
> > Despite being somewhat limited to a combination of existing methods, this paper still makes an important contribution to LiDAR NVS. Considering the authors' efforts in the rebuttal, which addressed most of my concerns, I am willing to increase my score.

---

> > > ### Author Response · Authors · 2024-11-25
> > >
> > > We appreciate the reviewer's time for reviewing and thanks again for the valuable comments and the positive score!

---

### Meta-Review · Area_Chair_1NfV · 2024-12-22

**Metareview:**

The paper introduces GS-LiDAR, a framework for LiDAR novel view synthesis (NVS). The key idea is to leverage 2D Gaussian primitives and periodic vibration properties to effectively reconstruct static and dynamic elements in driving scenarios. Specifically, it allows one to sidestep the limitations of previous NeRF-based hashgrid approach, ie, symmetrical structure and fixed resolution. The major concern of the reviewers is that the proposed method is a simple adoption of existing techniques without much innovation (eg, replacing NeRF with Gaussian Splatting). Although this replacement improves training and rendering speed compared to LIDAR4D, these advantages are fundamentally linked to the characteristics of Gaussian Splatting. The reviewers are thus on the fence both pre- and post-rebuttal. The ACs totally understand the dilemma of the reviewers. After extensive discussion, the ACs agree that while the approach is a combination of many existing components, the whole framework still push forward the frontier of LiDAR-NVS a bit. The experimental results are also rather comprehensive. The ACs thus recommend acceptance.

**Additional Comments On Reviewer Discussion:**

The reviewers are concerned about the novelty and implementation details of the proposed approach. The authors provided thorough response during the rebuttal.

---

### Decision · Program_Chairs · 2025-01-22

Accept (Poster)